# A multilevel analysis of the predictors of health facility delivery in Ghana: Evidence from the 2014 Demographic and Health Survey

**Justice Moses K. Aheto**[1,2,3]*, **Tracy Gates**[3], **Isaac Tetteh**[4], **Rahmatu Babah**[1]

**1** Department of Biostatistics, School of Public Health, College of Health Sciences, University of Ghana, Accra, Ghana, **2** WorldPop, School of Geography and Environmental Science, University of Southampton, Southampton, United Kingdom, **3** College of Public Health, University of South Florida, Tampa, Florida, United States of America, **4** Department of Community Health, University of Ghana Medical School, Korle-Bu, University of Ghana, Accra, Ghana

* justiceaheto@yahoo.com, jmkaheto@ug.edu.gh

**Data Availability Statement:** Data is freely available upon making official request to MEASURE DHS Team through the website at http://dhsprogram.com/data/available-datasets.cfm. A

## Abstract

Health facility delivery has the potential to improve birth and general health outcomes for both newborns and mothers. Regrettably, not all mothers, especially in low-and-middle income countries like Ghana deliver at health facilities, and mostly under unhygienic conditions. Using data from the 2014 Ghana Demographic and Health Survey, we fitted both weighted single-level and random intercept multilevel binary logistic regression models to analyse predictors of a health facility delivery among mothers aged 15–49 years and to quantify unobserved household and community differences in the likelihood of health facility delivery. We analysed data on 4202 mothers residing in 3936 households and 427 communities. Of the 4202 mothers who delivered, 3031 (75.3%—weighted and 72.1%—unweighted) delivered at the health facility. Substantial unobserved household only (Median Odds Ratio (MOR) = 5.1) and household conditional on community (MOR = 4.7) level differences in the likelihood of health facility delivery were found. Mothers aged 25–34 (aOR = 1.4, 95%CI: 1.0–2.1) and 35–44 (aOR = 2.9, 95%CI: 1.7–4.8), mothers with at least a secondary education (aOR = 2.7, 95%CI: 1.7–4.1), with health insurance coverage (aOR = 1.6, 95%CI: 1.2–2.2) and from richer/richest households (aOR = 8.3, 95%CI: 3.6–19.1) and with piped water (aOR = 1.5, 95%CI: 1.1–2.1) had increased odds of health facility delivery. Mothers residing in rural areas (aOR = 0.3, 95%CI: 0.2–0.5) and with no religion (aOR = 0.5, 95%CI: 0.3–1.0) and traditional religion (aOR = 0.2, 95%CI: 0.1–0.6), who reported not wanting to go to health facilities alone as a big problem (aOR = 0.5, 95%CI: 0.3–0.8) and having a parity of 2 (aOR = 0.4, 95%CI: 0.3–0.7), 3 (aOR = 0.3, 95%CI: 0.2–0.6) and ≥4 (aOR = 0.3, 95%CI: 0.1–0.5) had reduced odds of health facility delivery. Our predictive model showed outstanding predictive power of 96%. The study highlights the need for improved healthcare seeking behaviours, maternal education and household wealth, and bridge the urban-rural gaps to improve maternal and newborn health outcomes.

simple registration is required to freely access and download the data files for use.

**Funding:** The authors received no specific funding for this work.

**Competing interests:** The authors have declared that no competing interests exist

## Introduction

Though notable global improvements have been made over the last twenty years, maternal and neonatal mortality remain persistent issues faced primarily by low-and middle-income countries (LMICs), particularly those in sub-Saharan Africa (SSA). Maternal mortality refers to female deaths due to, or aggravated by, pregnancy related causes during pregnancy or within 42 days of termination; neonatal mortality consists of newborn deaths occurring within the first 28 days of life [1,2]. In 2016, maternal mortality was the second leading cause of death among women of reproductive age globally, 99% of which was accounted for in low-income countries, and more than half of which occurred in SSA [2,3]. SSA is also plagued by the highest neonatal mortality rate in the world with 28 deaths per 1,000 live births in 2018 [1]. As a country in SSA, Ghana is no exception. The Ghanaian maternal mortality ratio in 2019 was 310 deaths per 100,000 live births, and the Ghanaian neonatal mortality ratio in 2019 was 25 deaths per 1,000 live births, accounting for 68% of infant mortality and 48% of under-five mortality [4].

The national and international community have prioritized reducing maternal and neonatal mortality to less than 70 maternal deaths per 100,000 live births and 12 neonatal deaths per 1,000 live births by 2030 as outlined in the Sustainable Development Goals (SDGs) adopted by all United Nations (UN) Member States in pursuit of ending poverty, improving lives, and protecting the planet [5,6]. SDG 3 focuses on good health and well-being of everyone, everywhere. In pursuit of this goal, SDG target 3.7 prioritizes universal access to sexual and reproductive healthcare services, including health facility deliveries. Access and utilization of health facility deliveries is further prioritized in SDG indicator 3.1.2 which emphasizes the importance of the presence of skilled birth attendants during delivery (SBAs), the single most effective intervention for preventing maternal and neonatal deaths [6–14]. SBAs, as defined in a joint statement by the World Health Organization (WHO) along with six other United Nations (UN) and international associations in 2018, are educated and competent maternal and newborn health professionals trained and regulated to national and international standards [15].

The large majority of SBA attended deliveries occur in health facilities [10,12]. In the 2014 Ghana Demographic and Health Survey (GDHS) [16], we found that 99.6% of health facility deliveries were attended by a doctor, nurse, midwife, or community health officer compared to only 4.0% of deliveries occurring at other locations (e.g., home). Though some health facilities may be lacking in certain regards, it is well established that utilizing health facilities for delivery has the potential to reduce maternal and newborn morbidity and mortality and other complications during and immediately after delivery thereby maximising the survival chances of mothers and their newborns [13,16–22].

Three quarters of maternal deaths occur due to complications during pregnancy and childbirth, including obstructed labor, retained placenta, postpartum hemorrhage, sepsis, eclampsia, and complications of unsafe abortions, all of which are largely preventable with the presence of an SBA during deliveries at health facilities [2,10,14,19,23,24]. Similarly, in health facilities SBAs can provide crucial support to newborns during the critical 24-hour period following birth during which 43.6% of neonatal deaths occur, most of which are preventable and treatable [8].

Though overall Ghanaian utilization of skilled birth attendance improved from 59% in 2008 to 79% in 2019, these improvements were not uniform; those in urban areas utilized supervised delivery services 90% of the time, while those in rural areas utilized supervised delivery services only 59% of the time contributing to the majority of maternal deaths occurring in rural areas [2,4,12,19]. Previously identified barriers to giving birth in a health facility

or receipt of skilled birth attendance include long travel distances, poor socio-economic status, lack of health insurance coverage, poor antenatal care coverage or quality, advancing maternal age, increasing parity (birth order), low education level, low knowledge or experience of pregnancy complications, limited media exposure, lack of partner approval, religion, and perceived societal norms against health facility deliveries [9,11,12,14,19,25–28]. Ghana is committed to improving maternal and neonatal health outcomes as evidenced in their commitment to the Global Strategy for Women's, Children's, and Adolescents' Health (GSWCAH) and to the Global Network to improve the Quality of Care for Maternal and Newborn Health as well as their implementation of the community-based health planning and services (CHPS), the Safe Motherhood Initiative, the national health insurance scheme (NHIS), the free delivery policy, High Impact Rapid Delivery (HIRD), and the Emergency Obstetric and Neonatal Care program [2,9,12,25,29]. Though maternal and child healthcare, and delivery services are free, they are not universally accessible due to various informal costs and socio-cultural and socio-demographic barriers [2,12]. These challenges have resulted in unequal advancements in maternal health and birth outcomes, and continuing disparities across the country [19].

Previous studies have assessed determinants of health facility deliveries using multivariable logistic regression models. However, no research has yet analysed the data utilizing multilevel regression modelling and Geospatial mapping approaches to further investigate grouped data and interactions across multiple levels, and how the observed proportion of health facility delivery differs geographically across the regions in Ghana [19,30–32]. Due to the underutilization of health facilities during childbirth and prevalent disparities across the country, it is critical to develop sophisticated but accurate models to predict and understand the predictors of health facility delivery to support sound evidence-based decision making among policymakers responsible for improving the health outcomes of women and their newborns. The purpose of this study is therefore to develop a multilevel predictive model to predict health facility delivery while simultaneously adjusting for maternal, household, and community factors.

## Materials and methods

### Study design

The sampling frame used for the 2014 Ghana Demographic and Health Survey (GDHS) was an updated frame based on the 2010 Ghana Population and Housing Census. A two-stage sample design was utilized with the first stage been the selection of sample points (i.e., clusters) made up of enumeration areas (EAs) in which a total of 427 clusters consisting of 216 urban and 211 rural were selected. This was followed by systematic random sampling through household listing exercise carried out in all the selected EAs from January–March 2014 as the second stage. To allow estimation of reliable key indicators at the national and regional levels, including rural urban communities, the two-stage design was utilized [16]. From each cluster, about 30 households from the list were randomly selected, resulting in a total of 12831 households. Three separate questionnaires namely household, men, and women questionnaires were administered in the study. In the women's questionnaire, data on maternal and child health, maternal health seeking behaviour, awareness and use of family planning methods, breastfeeding practices, nutritional status of women and children, fertility preferences, childhood mortality and domestic violence were among some variables collected.

### Data

The study utilized data from the nationally representative 2014 GDHS which is freely available online upon request at the DHS MEASURE Program website which is the most recent full DHS data available for Ghana [33]. The present study utilised data from the women's

questionnaire where eligible women were asked to report the place of delivery for each child born in the five years preceding the survey. Data on background characteristics of the women such as education, media exposure, reproductive history, age, religion, knowledge of uses and sources of family planning methods, antenatal care (ANC), delivery, postnatal care, new-born care, and husband's background characteristics were collected. We extracted data on 4202 women of reproductive age (i.e., 15–49 years) residing in 3936 households and 427 clusters (i.e., communities) in the women's data set for this study. Detailed description of the methods and procedures used in the 2014 GDHS is published elsewhere [16].

## Outcome variables

The primary outcome variable of interest was health facility delivery utilization measured by place of delivery [16] which we classified as health facility delivery coded as 1 and non-health facility delivery coded as 0.

## Independent variables

The present study considered several covariates informed by the available literature on predictors of health facility delivery or skilled birth attendance [9,11,12,14,19,25,30,32,34,35]. These include maternal education (categorized as no education, primary and secondary/higher), getting permission to go for treatment (categorized as big problem and not a big problem), getting money needed for treatment (categorized as big problem and not a big problem), distance to health facility (categorized as big problem and not a big problem), not wanting to go alone (categorized as big problem and not a big problem), maternal age (categorized as 15–24, 25–34, 35–44 and 45–49), health insurance (no and yes), household wealth (categorized as poorer/poorest, middle and richer/richest), place of residence (categorized as rural and urban), type of water in household (categorized as unpiped and piped), toilet facility in household (categorized as non-flush and flush), respondent's religion (categorized as Christian, Islam, traditionalist/spiritualist and no religion), frequency of listening to radio (categorized as not at all, less than once a week and at least once a week), frequency of watching TV (categorized as not at all, less than once a week and at least once a week), and parity (categorized as 1, 2, 3, and $\geq 4$). The categories of some independent variables presented here such as education, wealth index, and type of water were combined to allow for sufficient sample size for each category for the variables to obtain reliable parameter estimates from the fitted model. The study also considered other variables (e.g., occupation, marital status, antenatal care visits, partner's education) but these were removed during model selection using both backward elimination and forward selection procedures and were not retained in the final model. This model selection process is critical to identifying sets of predictors that provide a good fit to the data in this study because it is possible that the wholesome inclusion of all these identified predictors informed by the available literature could lead to the overfitting of the multivariable logistic regression models and its resultant misleading estimated model parameters.

## Statistical analysis

We summarize the distribution of the selected background characteristics of the respondents and performed further analyses to examine individual, household, and community-level predictors of health facility delivery utilization and investigated residual household and community-level effects on health facility delivery. For both descriptive and inferential analyses, we accounted for the sampling weights in the DHS survey. We applied both single-level (level 1—women) and multilevel (level 2 –households and level 3—community) binary logistic regression models to analyse data on a total of 4201 mothers residing in 3936 households and 427

communities with complete measurements on health facility delivery as well as complete measurements on potential predictors considered in the study. The minimum number of mothers living in household was one (1) and maximum was four (4). For the cluster (i.e., community), the minimum and the maximum number of women per cluster was 1 and 33 respectively. We extend the single level (level 1) binary logistic regression model to the multilevel binary logistic regression model (levels 2 and 3) to account for the hierarchical structure of the GDHS dataset where we have individual mothers nested within households which are nested within communities. Specifically, the multilevel binary logistic regression model was applied to examine possible variations in utilization of health facility delivery among mothers across households and communities while simultaneously identifying potential predictors. This multilevel modelling study [36] placed an importance on household and community-level differences in the odds of health facility delivery among women and the extent of nesting of health facility delivery within a household and community which cannot be achieved through the traditional single level binary logistic regression model. Ignoring the nested structure of the data could result in wrong estimation of model parameters, especially the standard errors which could lead to spurious statistical significance, a wrong conclusion and misleading policy decisions [37].

## Model formulation

Presented here is the multilevel binary logistic regression model formulation in which we allowed for clustering/nesting (i.e., individuals nested within households) in the data. Let $\pi_{ij}$ be the probability that individual $i$ from household $j$ delivered at health facility, $P_{ij} = \left(\frac{\pi_{ij}}{1-\pi_{ij}}\right)$ is the odds of health facility delivery by individual $i$ living in household $j$. Our model formulation is given as:

$log\left(\frac{\pi_{ij}}{1-\pi_{ij}}\right) = \alpha + \boldsymbol{d}(\boldsymbol{x_{ij}})'\beta + h_{0j}$, where $\alpha$ is the overall mean probability of health facility delivery shared by all women across households, $d(.)$ is a vector of predictors, $\beta$ is a vector of regression coefficients, $h_{0j}$ is the unobserved household-level effect for household $j$ which is assumed to follow a normal distribution with mean zero (0) and variance $\sigma_h^2$. The individual-level residual $e_{ij}$ is assumed to follow standard logistic distribution with zero mean and variance $\pi^2/3$, where $\pi \approx 3.14$ [38].

We extend the household-level model to a three-level model by allowing for random effect at the cluster (i.e., community) level. Our three-level (community) model formulation is given as:

$log\left(\frac{\pi_{ijk}}{1-\pi_{ijk}}\right) = \alpha + \boldsymbol{d}(\boldsymbol{x_{ijk}})'\beta + h_{0jk} + c_{0k}$, where $\alpha$ is the overall mean probability of health facility delivery shared by all women across households and communities, $h_{0jk}$ is the unobserved effect for household $j$ in community $k$ and $c_{0k}$ is the unobserved effect for community $k$ which follows a normal distribution with mean zero (0) and variance $\sigma_c^2$. $\boldsymbol{d}(.)$ is a vector of predictors that can be defined at the mothers, household, and/or community levels. We implement the multilevel models under the mean-variance adaptive Gauss-Hermite quadrature integration approach for likelihood approximation [39].

We computed the variance partitioning coefficient (VPC) [40] to quantify the proportion of total variation attributable to within-household and within-community differences. For within-household (2-level) differences, we have $VPC_h = \frac{\sigma_h^2}{\sigma_h^2 + \frac{\pi^2}{3}}$ x 100 and household within-community (3-level) differences given as $VPC_c = \frac{\sigma_c^2 + \sigma_h^2}{\sigma_c^2 + \sigma_h^2 + \frac{\pi^2}{3}}$ x 100. The quantity $VPC_c$ measures the percentage of residual variation due to differences between households in different communities.

Due to lack of physical interpretation of the VPC, we converted the VPC to median odds ratio (MOR) as a measure of household-level heterogeneity or variation. The MOR is the median odds ratio between a woman in a household or community with a higher probability of health facility delivery and a woman in another household or community with a lower probability of health facility delivery given similar woman level characteristics [41–44]. The MOR is much easier to interpret and understand as it is expressed in terms of inter-household or inter-community heterogeneity on the usual odds ratio scale based on which the effects of the predictors are also interpreted [42,43,45]. It is a measure of spread on the odds ratio scale and provides information about the average variance between two random households or communities. With our household-level only variance as $\sigma_h^2$, the MOR is estimated as:

$MOR_h = exp(\sqrt{2 \times \sigma_h^2} \times \varphi^{-1}(0.75)) \cong exp(0.945 \times \sqrt{\sigma_h^2})$ and for our household within-community-level variance $\sigma_h^2$ and $\sigma_c^2$, we have:

$$MOR_c = exp(\sqrt{2 \times (\sigma_c^2 + \sigma_h^2)} \times \varphi^{-1}(0.75)) \cong exp(0.945 \times \sqrt{\sigma_c^2 + \sigma_h^2})$$

Maximum likelihood estimation procedure was used to obtain model parameters. Among the covariance structures, the independent structure provided a good fit to the data in the multilevel binary logistic regression models. Akaike Information Criterion (AIC) was used to select the model that fits the data better. To examine multicollinearity in our single-level multivariable model, we employed the generalised variance inflation factor (GVIF), and a value below 10 was considered acceptable (i.e., no noticeable multicollinearity) [44,46,47].

We investigated the predictive performance of our fitted models using Area Under Receiver Operating Characteristics (AUROC) curve. All the analyses were performed using Stata version 16.1 [48]. To declare statistical significance, we considered a p-value < 0.05.

## Ethics approval and consent to participate

The 2014 GDHS protocol was reviewed and approved by the Ghana Health Service Ethical Review Committee and the Institutional Review Board of ICF International. A written consent to participate was obtained from all participants, and for minors, a parent or guardian provided consent prior to participation [16]. Though this study did not require ethical approval because the GDHS dataset is publicly available and this study was granted permission to use the data by the MEASURE DHS Program, it is worth mentioning that ethical clearance procedures were strictly adhered to before, during and after the GDHS to ensure protection of human rights as required by the US Department of Health and Human Services. Detailed information on DHS data and ethical standards are publicly available at http://goo.gl/ny8T6X.

## Results

### Background characteristics of study participants

Presented in Table 1 is the summary of background characteristics of mothers surveyed in the 2014 GDHS. Out of 4202 mothers who delivered, 3031 (75.3%—weighted proportion and 72.1%—unweighted proportion) delivered in a health facility. Those aged 25–34 were the majority (47.1%) while those aged 45–49 were the minority (3.5%). Majority (46.4%) had secondary/higher education and 2457 (58.5%) of them reside in rural communities. Christians are the majority religious group (71.7%) while traditional/spiritualist were in minority (3.7%). Mothers from the poorer/poorest households were in majority (52.2%) while 3920 (93.3%) of the mothers felt that getting permission for treatment was not a big problem. Getting money for treatment was a big problem for about 43.2% of the mothers. The distance to a health

**Table 1. Distribution of selected background characteristics of respondents by place of delivery (N = 4202).**

| Characteristics | Frequency n(%) | Non health facility (n(%)¥) | Health facility (n(%)¥) | P-value |
|---|---|---|---|---|
| **Age group (Years)** | | | | 0.005 |
| 15–24 | 891 (21.2) | 238 (22.2) | 653 (20.9) | |
| 25–34 | 1981 (47.1) | 538 (46.1) | 1443 (47.9) | |
| 35–44 | 1183 (28.2) | 331 (26.9) | 852 (28.8) | |
| 45–49 | 147 (3.5) | 64 (4.8) | 83 (2.4) | |
| **Educational level** | | | | <0.001 |
| No education | 1399 (33.3) | 644 (48.3) | 755 (19.0) | |
| Primary | 851 (20.3) | 265 (24.6) | 586 (18.0) | |
| Secondary/higher | 1952 (46.4) | 262 (27.1) | 1690 (63.0) | |
| **Place of residence** | | | | <0.001 |
| Urban | 1745 (41.5) | 180 (16.0) | 1565 (56.4) | |
| Rural | 2457 (58.5) | 991 (84.0) | 1466 (43.6) | |
| **Religion** | | | | <0.001 |
| Christian | 3013 (71.7) | 726 (64.9) | 2287 (80.1) | |
| Islam | 865 (20.6) | 236 (18.3) | 629 (16.2) | |
| Traditional/spiritualist | 154 (3.7) | 114 (9.1) | 40 (1.0) | |
| No religion | 170 (4.1) | 95 (7.8) | 75 (2.6) | |
| **Household wealth** | | | | <0.001 |
| Poorer/poorest | 2194 (52.2) | 940 (74.7) | 1254 (30.3) | |
| Middle | 792 (18.9) | 177 (18.7) | 615 (20.4) | |
| Richer/richest | 1216 (28.9) | 54 (6.6) | 1162 (49.4) | |
| **Getting permission to go for treatment** | | | | 0.023 |
| Big problem | 281 (6.7) | 100 (7.4) | 181 (5.2) | |
| Not big problem | 3920 (93.3) | 1071 (92.6) | 2849 (94.8) | |
| **Getting money needed for treatment** | | | | <0.001 |
| Big problem | 2023 (48.2) | 724 (57.1) | 1299 (39.5) | |
| Not big problem | 2178 (51.8) | 447 (42.9) | 1731 (60.5) | |
| **Distance to health facility** | | | | <0.001 |
| Big problem | 1296 (30.9) | 527 (41.2) | 769 (22.0) | |
| Not big problem | 2905 (69.1) | 644 (58.8) | 2261 (78.0) | |
| **Not wanting to go health facility alone** | | | | <0.001 |
| Big problem | 622 (14.8) | 268 (22.5) | 354 (11.2) | |
| Not big problem | 3579 (85.2) | 903 (77.5) | 2676 (88.8) | |
| **Has health insurance** | | | | <0.001 |
| No | 1299 (30.9) | 475 (42.5) | 824 (30.4) | |
| Yes | 2902 (69.1) | 696 (57.5) | 2206 (69.6) | |
| **Type of water in household** | | | | <0.001 |
| Unpiped | 3023 (71.9) | 966 (81.9) | 2057 (67.9) | |
| Piped | 1179 (28.1) | 205 (18.1) | 974 (32.1) | |
| **Toilet facility in household** | | | | <0.001 |
| Non-flush | 3634 (86.5) | 1442 (96.3) | 2492 (75.4) | |
| Flush | 568 (13.5) | 29 (3.7) | 539 (24.6) | |
| **Frequency of listening to radio** | | | | <0.001 |
| Not at all | 806 (19.2) | 345 (26.5) | 461 (14.7) | |
| Less than once a week | 1290 (30.7) | 381 (35.6) | 909 (31.3) | |
| At least once a week | 2106 (50.1) | 445 (37.9) | 1661 (54.0) | |

*(Continued)*

**Table 1.** (Continued)

| Characteristics | Frequency n(%) | Non health facility (n(%)¥) | Health facility (n(%)¥) | P-value |
|---|---|---|---|---|
| **Frequency of watching TV** | | | | <0.001 |
| Not at all | 1546 (36.8) | 666 (49.2) | 880 (22.0) | |
| Less than once a week | 924 (22.0) | 235 (23.7) | 689 (26.1) | |
| At least once a week | 1732 (41.2) | 270 (27.1) | 1462 (51.9) | |
| **Birth order number (parity)** | | | | <0.001 |
| 1 | 900 (21.4) | 140 (12.3) | 760 (25.8) | |
| 2 | 817 (19.4) | 178 (16.7) | 639 (21.4) | |
| 3 | 713 (17.0) | 188 (16.6) | 525 (18.1) | |
| ≥4 | 1772 (42.2) | 665 (54.5) | 1107 (34.7) | |

Note: The p-values are based on Pearson Chi-squared test of independence; n: Unweighted frequency; ¥: Weighted percentage.

facility was not a big problem for 2905 (69.1%) of the mothers, and mothers who felt it was a big problem to go to the health facility alone were 622 (14.8%). About 1299 (30.9%) of the mothers did not have health insurance coverage while majority of them lived in households without piped water (71.9%) and flush toilets (86.5%). Majority of the mothers listened to radio (50.1%) and watched TV (42.2%) at least once a week and those with birth order number of 4 or more were in majority (42.2%) while those with birth order number of 3 were in minority (17.0%) (Table 1).

All the selected background characteristics presented in Table 1 were significantly associated with health facility delivery status and were all included in the single and multilevel regression models.

## Model selection and predictive ability of the fitted models

We set off by fitting a single-level multivariable model where we adjusted for the selected predictors (Model 1). We then extended model 1 to account for the household level hierarchy only (Model 2) and community level hierarchy only (Model 3) using the same set of predictors used in model 1. Finally, we extended model 2 to allow both the household and community-level hierarchies simultaneously (Model 4) while simultaneously adjusting for the same predictors in model 2. Based on both the AIC and Log Pseudo-Likelihood statistics, our preferred model is model 4 which accounted for the individual mothers, household, and community level hierarchical structure of the GDHS data set (Table 2).

**Table 2. Model selection, and evaluating the predictive performance of the fitted models for health facility delivery.**

| | Model selection Model predictive performance | | | | | |
|---|---|---|---|---|---|---|
| **Model** | **AIC** | **Log (PL)** | **N** | **AUROC (%)** | **SE** | **95% CI** |
| Model 1 | 3437.9 | -1692.9 | 4201 | 80.4 | 0.0070 | 79.0, 81.8 |
| Model 2 | 3307.2 | -1626.6 | 4201 | 96.0 | 0.0031 | 95.4, 96.6 |
| Model 3 | 3268.4 | -1607.2 | 4201 | 88.4 | 0.0055 | 87.3, 89.4 |
| Model 4 | 3267.7 | -1605.8 | 4201 | 95.9 | 0.0030 | 95.3, 96.5 |

AIC: Akaike Information Criterion, PL: Pseudo-Likelihood, AUROC: Area Under the Receiver Operating Characteristics curve, SE: Prediction Standard Error, CI: Confidence Interval.

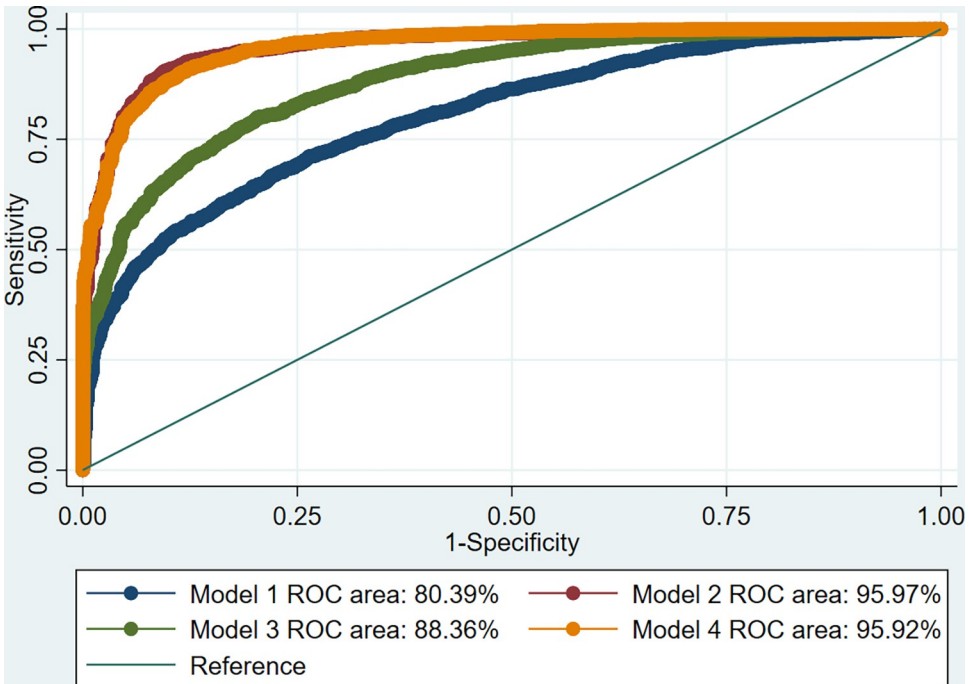

**Fig 1. AUROC curve of single- and multilevel binary logistic regression models for predicting health facility delivery among mothers in Ghana.**

To examine the predictive ability/performance of our fitted models, we employed AUROC curve, and the results were presented in Table 2 and Fig 1. Our results showed that model 2 (i.e., household level only model– 2-level) (AUROC value = 96.0%) had slightly better predictive performance than model 4 (i.e., household and community level– 3-level) (AUROC value = 95.9%) even though the difference is negligible, and it had relatively higher prediction standard error (0.0031) compared to model 4 (0.0030) (Table 2). Both models showed an outstanding predictive accuracy of predicting health facility delivery among mothers in Ghana. Considering the model selection statistics (i.e., AIC and Log Pseudo-Likelihood) which favored model 4, and model predictive performance statistics (prediction standard error—SE), our preferred model for predicting health facility delivery by mothers is model 4 (Table 2 and Fig 1).

### Predictors of health facility delivery

**Univariable analyses.** The univariable (unadjusted) single level binary logistic regression analyses were presented in Table 3. Significant predictors of health facility delivery were age, education level, place of residence, religion, permission to go for treatment, money for treatment, distance to a health facility, not wanting to go health facility alone, health insurance, household wealth, type of water in the household, toilet facility in household, frequency of listening to radio, frequency of watching TV, and parity.

**Single-level multivariable analyses.** In the single level multivariable model, age, education level, place of residence, religion, not wanting to go to a health facility alone, health insurance coverage, household wealth, type of water in the household, frequency of listening to radio, and parity were found to be significantly and independently predictive of health facility delivery. Getting permission to go for treatment, getting money needed for treatment, distance

**Table 3. Predictors of health facility delivery from the weighted single-level and multilevel binary logistic regression models.**

| Characteristics | Single level | | 2 Level | | 3 level |
| --- | --- | --- | --- | --- | --- |
| | uOR (95% CI) | Model 1 aOR (95% CI) | Model 2 aOR (95% CI) | Model 3 aOR (95% CI) | Model 4 aOR (95% CI) |
| **Age group (Years)** | | | | | |
| 15–24 | Ref | ref | ref | ref | ref |
| 25–34 | 1.1 (0.9–1.4) | 1.4 (1.0–1.9) | 1.5 (1.0–2.2) | 1.4 (1.0–1.8) | 1.4 (1.0–2.1) |
| 35–44 | 1.1 (0.9–1.5) | 2.3 (1.6–3.4) | 3.1 (1.9–5.0) | 2.5 (1.7–3.6) | 2.9 (1.7–4.8) |
| 45–49 | 0.5 (0.3–0.8) | 2.0 (1.2–3.5) | 2.6 (1.3–5.3) | 1.8 (1.0–3.2) | 2.0 (1.0–4.1) |
| **Educational level** | | | | | |
| No education | ref | ref | ref | ref | ref |
| Primary | 1.9 (1.5–2.3) | 1.4 (1.1–1.8) | 1.6 (1.1–2.2) | 1.3 (1.0–1.9) | 1.4 (1.0–2.2) |
| Secondary/higher | 5.9 (5.0–7.3) | 2.2 (1.7–2.9) | 3.2 (2.3–4.4) | 2.3 (1.7–3.0) | 2.7 (1.7–4.1) |
| **place of residence** | | | | | |
| Urban | ref | ref | ref | ref | ref |
| Rural | 0.2 (0.1–0.2) | 0.5 (0.4–0.6) | 0.4 (0.2–0.5) | 0.4 (0.3–0.5) | 0.3 (0.2–0.5) |
| **Religion** | | | | | |
| Christian | ref | ref | ref | ref | ref |
| Islam | 0.7 (0.6–0.9) | 1.2 (0.9–1.5) | 1.2 (0.8–1.6) | 1.1 (0.8–1.7) | 1.2 (0.8–1.8) |
| Traditional/spiritualist | 0.1 (0.1–0.1) | 0.3 (0.2–0.4) | 0.2 (0.1–0.3) | 0.3 (0.1–0.6) | 0.2 (0.1–0.6) |
| No religion | 0.3 (0.2–0.4) | 0.5 (0.4–0.8) | 0.3 (0.2–0.6) | 0.6 (0.4–1.0) | 0.5 (0.3–1.0) |
| **Getting permission to go for treatment** | | | | | |
| Not big problem | ref | ref | ref | ref | ref |
| Big problem | 0.7 (0.5–0.9) | 1.3 (0.9–1.8) | 1.3 (0.8–2.1) | 1.2 (0.8–1.9) | 1.3 (0.8–2.1) |
| **Getting money needed for treatment** | | | | | |
| Not big problem | ref | ref | ref | ref | ref |
| Big problem | 0.5 (0.4–0.6) | 1.1 (0.9–1.4) | 1.0 (0.7–1.3) | 1.0 (0.8–1.4) | 1.0 (0.7–1.5) |
| **Distance to health facility** | | | | | |
| Not big problem | ref | ref | ref | ref | ref |
| Big problem | 0.4 (0.3–0.5) | 0.9 (0.7–1.1) | 0.8 (0.6–1.1) | 1.1 (0.8–1.4) | 1.1 (0.8–1.5) |
| **Not wanting to go health facility alone** | | | | | |
| Not big problem | ref | ref | ref | ref | ref |
| Big problem | 0.4 (0.4–0.5) | 0.6 (0.5–0.8) | 0.5 (0.4–0.8) | 0.6 (0.4–0.8) | 0.5 (0.3–0.8) |
| **Has health insurance** | | | | | |
| No | ref | ref | ref | ref | ref |
| Yes | 1.7 (1.4–2.0) | 1.6 (1.3–1.9) | 2.0 (1.5–2.5) | 1.5 (1.2–1.9) | 1.6 (1.2–2.2) |
| **Household wealth** | | | | | |
| Poorer/poorest | ref | ref | ref | ref | ref |
| Middle | 2.7 (2.1–3.4) | 1.4 (1.0–1.8) | 1.7 (1.2–2.4) | 1.4 (1.0–1.9) | 1.5 (1.0–2.2) |
| Richer/richest | 18.5 (13.3–25.6) | 5.3 (3.5–8.0) | 9.0 (5.3–15.2) | 6.0 (3.3–11.1) | 8.3 (3.6–19.1) |
| **Type of water in household** | | | | | |
| Unpiped | ref | ref | ref | ref | ref |
| Piped | 2.2 (1.8–2.6) | 1.4 (1.1–1.7) | 1.4 (1.0–2.0) | 1.4 (1.0–1.9) | 1.5 (1.1–2.1) |
| **Toilet facility in household** | | | | | |
| Non-flush | ref | ref | ref | ref | ref |
| Flush | 8.5 (5.5–13.0) | 1.3 (0.7–2.1) | 1.3 (0.7–2.4) | 1.1 (0.6–2.0) | 1.1 (0.5–2.2) |
| **Frequency of listening to radio** | | | | | |
| Not at all | ref | ref | ref | ref | ref |
| Less than once a week | 1.6 (1.3–2.0) | 1.0 (0.8–1.3) | 1.1 (0.8–1.6) | 1.0 (0.7–1.4) | 1.0 (0.7–1.4) |

*(Continued)*

**Table 3.** (Continued)

| Characteristics | Single level | | 2 Level | | 3 level |
|---|---|---|---|---|---|
| | uOR (95% CI) | Model 1 aOR (95% CI) | Model 2 aOR (95% CI) | Model 3 aOR (95% CI) | Model 4 aOR (95% CI) |
| At least once a week | 2.6 (2.1–3.2) | 1.4 (1.1–1.7) | 1.7 (1.2–2.4) | 1.4 (0.9–1.8) | 1.3 (0.8–2.0) |
| **Frequency of watching TV** | | | | | |
| Not at all | ref | ref | ref | ref | Ref |
| Less than once a week | 2.5 (2.0–3.1) | 0.9 (0.7–1.2) | 0.8 (0.5–1.1) | 1.0 (0.7–1.3) | 1.0 (0.7–1.4) |
| At least once a week | 4.3 (3.5–5.2) | 0.9 (0.7–1.1) | 0.8 (0.6–1.2) | 0.9 (0.6–1.2) | 0.9 (0.6–1.3) |
| **Birth order number (parity)** | | | | | |
| 1 | ref | ref | ref | ref | ref |
| 2 | 0.6 (0.5–0.8) | 0.5 (0.4–0.8) | 0.5 (0.3–0.7) | 0.5 (0.3–7) | 0.4 (0.3–0.7) |
| 3 | 0.5 (0.4–0.7) | 0.5 (0.3–0.7) | 0.3 (0.2–0.6) | 0.4 (0.3–0.6) | 0.3 (0.2–0.6) |
| 4+ | 0.3 (0.2–0.4) | 0.4 (0.2–0.5) | 0.2 (0.1–0.4) | 0.3 (0.2–0.5) | 0.3 (0.1–0.5) |
| **Variation analysis** | | | | | |
| **Random effect parameters Estimates** | | | | | **Estimates** |
| Individual-level variance | - | 3.3 | 3.3 | 3.3 | 3.3 |
| Household-level variance ($\sigma_h^2$) | - | - | 3.0 (2.2–4.1) | - | 1.3 (0.3–6.4) |
| Household-level VPC | - | - | 47.5% (39.8–55.4) | - | 45.0% (23.1–69.1) |
| Household-level MOR | - | - | 5.1 | - | 4.7 |
| Community-level variance ($\sigma_c^2$) | - | - | - | 1.0 (0.7–1.3) | 1.4 (0.8–2.3) |
| Community-level VPC | - | - | - | 22.6% (17.6–28.7) | 22.9% (17.8–29.1) |
| Community-level MOR | - | - | - | 2.5 | 3.0 |

uOR: Unadjusted odds ratio; aOR: Adjusted odds ratio; CI: Confidence interval; ref: Reference category.

to a health facility, toilet facility in households, and frequency of watching TV were no longer statistically significant in the single-level multivariable model (Table 3, Model 1).

**Multilevel multivariable binary logistic regression analyses.** In this section, we presented the results for our final model (i.e., Model 4–3-level) presented in Table 3. Significant predictors of health facility delivery in the three-level multilevel model included age, educational level, place of residence, religion, not wanting to go to a health facility alone, health insurance coverage, household wealth, type of water in the household, and parity. Mothers aged 25–34 (aOR = 1.4, 95%CI: 1.0–2.1) and 35–44 (aOR = 2.9, 95%CI: 1.7–4.8), years had higher odds of delivering in a health facility compared to those aged 15–24 years. Mothers with at least a secondary education (aOR = 2.7, 95%CI: 1.7–4.1) had increased odds of delivering at health facilities compared to mothers with no formal education. Mothers in rural areas (aOR = 0.3, 95%CI: 0.2–0.5) had reduced odds of delivering at health facilities compared to their counterparts residing in urban areas. Mothers with no religion (aOR = 0.5, 95%CI: 0.3–1.0) and belonging to traditional religion (aOR = 0.2, 95%CI: 0.1–0.6) had lower odds of delivering at health facilities compared to those belonging to the Christian religion. Mothers who reported not wanting to go to health facilities alone as a big problem (aOR = 0.5, 95%CI: 0.3–0.8) had lower odds of delivering at health facility compared to their counterparts who reported that is not a big problem.

Mothers with health insurance coverage (aOR = 1.6, 95%CI: 1.2–2.2) had higher odds of health facility delivery compared to those without health insurance. Mothers from richer/richest households (aOR = 8.3, 95%CI: 3.6–19.1) had higher odds of health facility delivery compared to mothers from poorer/poorest households. Also, mothers who lived in households

with piped water (aOR = 1.5, 95%CI: 1.1–2.1) had increased odds of health facility delivery compared to mothers residing in households without piped water. Mothers with parity of 2 (aOR = 0.4, 95%CI: 0.3–0.7), 3 (aOR = 0.3, 95%CI: 0.2–0.6) and 4 or more (aOR = 0.3, 95%CI: 0.1–0.5) had reduced odds of a health facility delivery compared to mothers with a parity of 1 (Table 3 Model 4).

Substantial residual household level differences in different communities were observed in the fitted models for health facility delivery (Table 3). Specifically, about 45% of differences in the probability of health facility delivery could be attributable to households in different communities with associated MOR = 4.7. The MORs of 5.1 (households only), 2.5 (community 2-level only), and 3.0 (community 3-level) are at least two times higher than the reference value (MOR = 1), an indication of substantial household and community level differences in the probability of health facility delivery among mothers in Ghana.

## Discussion

This study set out to develop a multilevel predictive model to identify predictors of a health facility delivery. The focus of the study is to quantify unobserved household and community level residual effects on a health facility delivery by mothers, representing differences in the likelihood of health facility delivery across households and communities which cannot be explained by the considered predictors in the chosen model. The study found that significant number of Ghanaian mothers did not deliver in a health facility a finding consistent with previous studies. This is a threat to the goal of achieving universal coverage relating to health facility delivery, including SDG target 3.7 which is to guarantee universal access to sexual and reproductive health-care services [16,19,32,49].

This study was particularly interested in the unobserved household and community level differences in the likelihood of health facility delivery by mothers because the households and communities in which they reside could serve as a marker of shared hazards, opportunities, and resources available to them which in turn could determine their health seeking behaviour and health outcomes [37,47,50–52]. Substantial household and community level differences were observed in health facility delivery by women, suggesting that the utilization of health facility delivery by mothers differ significantly across households in different communities after adjusting for the set of predictors. In the household multilevel model, we found that a woman chosen at random from a household is at least 5 times likely to deliver in a health facility compared to another woman chosen randomly from another household. Similarly, the multilevel model containing both household and community-level random effects revealed that a woman chosen at random from a household in a community is nearly 5 times likely to deliver in a health facility compared to another woman chosen randomly from another household in a different community. This is an indication that any statistical model that ignores the hierarchical structure of the data will not be appropriate for the data, and will lead to spurious statistical significance and its associated misleading conclusions and policy decisions [37]. The presence of this substantial unobserved household and community level differences in the likelihood of health facility utilization could be attributable to cumulative effect of social, economic, and environmental/community factors related to households and geographical variations in health facility utilization among mothers which were not considered in our models. Policy and intervention strategies aimed at improving health facility delivery among women should target households and communities while further studies are warranted to identify reasons why some women residing in some households and communities had higher likelihood of delivering at health facility compared to their counterparts residing in other households and communities.

Furthermore, our multilevel predictive model was the first to show an outstanding predictive power based on AUROC value of 96%, the highest one can find in this area, indicating an excellent predictive ability of our model to correctly predict utilization of a health facility delivery in this population of mothers.

Critical factors identified to be independently and significantly predictive of the utilization of a health facility delivery include age, educational level, place of residence, religion, not wanting to go to a health facility alone, health insurance coverage, household wealth, source of water in household, and parity which broadly support findings from previous studies that examined utilization of health facility delivery and associated factors [19,26–28,30,35,53].

Generally, the effect sizes and the confidence intervals for the single-level (model 1) and the three-level (model 4) binary logistic regression models did not change that much except for household wealth (see rich/richer category for model 1 and model 4). Also, frequency of listening to radio was significant in model 1 but not significant in model 4, suggesting that the standard error was wrongly estimated in model 1, leading to the observed spurious statistical significance. However, changes in effect sizes are not the focus of this study as already presented. The focus is to quantify unobserved household and community differences in the likelihood of health facility delivery among women and the results (Table 3) confirmed that there were significant differences in the likelihood of health facility delivery across households in different communities in which the women reside, an important finding that cannot be achieved using the standard single-level binary logistic regression models.

To the best of our knowledge, our study is the first to identify improved source of water in households and mothers who reported big problems of not wanting to go to a health facility alone as predictive of health facility delivery utilization. A previous study only considered water source in health facilities under the health facility water, sanitation, and hygiene (WASH) program [26]. Mothers belonging to households with an improved water source had increased odds of utilizing health facility delivery compared to their counterparts with an unimproved water source. Mothers who reported not wanting to go to a health facility alone as a big problem were less likely to deliver at health facility.

Utilization of health facilities for deliveries can be improved through community outreach initiatives targeted at households aimed at educating people on mothers' right to access free maternal healthcare services, encouraging women to register for health insurance, and expanding health insurance coverage to low income, less educated, and rural communities. Socioeconomic improvement of households should be key as part of the overall strategy to address the problem of not delivering at a health facility with its attendant consequences of maternal and neonatal morbidity and mortality.

## Strengths and limitations

The utilization of nationally representative population-based data based on the standardized DHS data permits generalization of our findings to the population of Ghanaian mothers and mothers from similar populations outside Ghana. The utilization of a multilevel regression modelling approach, in addition to accounting for the sampling weights provides a more effective means of analysing nationally representative clustered data which allows for better prediction and correct inferences. Although validated questionnaires are used in the GDHS to ensure internal and external validity, our results should be interpreted and utilized with caution because of the nature of the GDHS data. Though widely respected, the GDHS survey is a cross-sectional, retrospective survey and therefore is unable to infer effect of causality and may be influenced by a recall or reporting bias. However, it is not expected that these biases would influence the observed estimates because the study was based on secondary data collected via

the robust validated DHS. Another limitation is the datedness of the data, but the 2014 GDHS is the most recent full DHS data available for Ghana that provides critical population level health estimates for the country, so the findings of this study is very relevant to decision-makers and other researchers interested in the utilization of multilevel modelling framework to analyse public health data. Just like any other study, this study could not measure or analyse all factors that can predict where mothers give birth, but this does not affect the overall interpretation and implication of our findings.

## Conclusion

Our analysis highlighted the need to target households and communities with public health policies and intervention strategies to increase uptake of health facility delivery in this population of women. This could be in the form of targeting women, their partners, family members and communities with health education and promotion messages related to the benefits of health facility delivery for mothers and the newborns. We recommend further studies to investigate and identify as-yet unidentified predictors such as health service quality and community level attitudes, actual distance from households to the nearest health facility, and healthcare policies among others not measured by the DHS program which might be responsible for the unexplained household and community level differences observed in the health facility delivery utilization among mothers to further inform targeted policies and intervention strategies. The study also identified sociodemographic, socioeconomic, and healthcare access factors that predict a health facility delivery, suggesting the need for a multifaceted and integrated approach to address inequalities in health facility delivery utilization.

## Acknowledgments

Special thanks go to DHS program for providing access to the datasets used in the current study.

## Author Contributions

**Conceptualization:** Justice Moses K. Aheto.

**Data curation:** Justice Moses K. Aheto, Tracy Gates, Isaac Tetteh, Rahmatu Babah.

**Formal analysis:** Justice Moses K. Aheto.

**Investigation:** Justice Moses K. Aheto, Tracy Gates, Isaac Tetteh, Rahmatu Babah.

**Methodology:** Justice Moses K. Aheto.

**Project administration:** Justice Moses K. Aheto.

**Resources:** Justice Moses K. Aheto.

**Software:** Justice Moses K. Aheto.

**Supervision:** Justice Moses K. Aheto.

**Validation:** Justice Moses K. Aheto.

**Visualization:** Justice Moses K. Aheto.

**Writing – original draft:** Justice Moses K. Aheto, Tracy Gates, Rahmatu Babah.

**Writing – review & editing:** Justice Moses K. Aheto, Tracy Gates, Isaac Tetteh, Rahmatu Babah.

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
