## [Decision Letter · Decision Letter 0]

2 May 2023

PGPH-D-22-01485

A multilevel predictive modelling and predictors of health facility delivery in Ghana: evidence from the 2014 Demographic and Health Survey

Dear Dr. Aheto,

Thank you for submitting your manuscript to PLOS Global Public Health. After careful consideration, we feel that it has merit but does not fully meet PLOS Global Public Health’s publication criteria as it currently stands. Therefore, we invite you to submit a revised version of the manuscript that addresses the points raised during the review process.

We look forward to receiving your revised manuscript.

Kind regards,

Bethany Hedt-Gauthier, PhD

Academic Editor

Journal Requirements:

1. Please provide additional details regarding participant consent. In the ethics statement, please ensure that you have specified what type you obtained (for instance, written or verbal, and if verbal, how it was documented and witnessed). If your study included minors, state whether you obtained consent from parents or guardians.

Additional Editor Comments (if provided):

Thank you for your submission, and we acknowledge the delays in review due to our challenges in finding suitable reviewers who agreed to this assignment. We have now met our target.

You will see detailed feedback from the reviewers. Of particular note are the concerns about the datedness of the data and the overlap with other work. The authors address some of these latter points in their paper, but this needs more discussion on the overlap and more emphasis on the unique contributions of this paper.

Reviewers' comments:

Reviewer's Responses to Questions

**Comments to the Author**

1. Does this manuscript meet PLOS Global Public Health’s publication criteria? Is the manuscript technically sound, and do the data support the conclusions? The manuscript must describe methodologically and ethically rigorous research with conclusions that are appropriately drawn based on the data presented.

Reviewer #1: Yes

Reviewer #2: Yes

Reviewer #3: Yes

2. Has the statistical analysis been performed appropriately and rigorously?

Reviewer #1: Yes

Reviewer #2: Yes

Reviewer #3: Yes

3. Have the authors made all data underlying the findings in their manuscript fully available (please refer to the Data Availability Statement at the start of the manuscript PDF file)?

Reviewer #1: Yes

Reviewer #2: Yes

Reviewer #3: Yes

4. Is the manuscript presented in an intelligible fashion and written in standard English?

Reviewer #1: Yes

Reviewer #2: Yes

Reviewer #3: Yes

5. Review Comments to the Author

Reviewer #1: This manuscript would be a helpful contribution to the knowledge of predictors of facility-based delivery for improved newborn and maternal outcomes in low- and middle-income countries. I hope the authors will find the comments below helpful as they revise the paper.

Title:

1) The title may need some improvement for easier reading and understanding. For example: “A multilevel analysis of the predictors of health facility delivery in Ghana: evidence from the 2014 Demographic and Health Survey”.

Abstract:

2) In order to help the reader of this paper, you should briefly describe what the Sustainable Development Goal 3 and its target 7 say.

Methods’ section:

3) “Independent variables”: you need to focus on defining your variables (i.e. how GDHS defines each variable and describe any other changes that you might have made to these variables to fit your analyses) and avoid to provide unnecessary details (e.g. numeric codes given to categorical variables) that may confuse readers.

4) It’s important that you cited a number of papers that previously studied determinants of facility-based delivery, however it is not clear which conceptual framework you adopted/adapted? There is also a number of potential predictor variables like antenatal care (ANC) services utilization, partner’s education and marital status that you didn’t include in your analysis, why?

5) For the “household wealth index” variable, explain the rationale of collapsing poorer/poorest and richer/richest categories and NOT using the original 5 categories variable as it is provided by GDHS?

6) Explain why you included in your model the “type of water” and “toilet facility” in the household variables together with the “wealth index” variable, while they are also used in the calculation of the wealth index as a composite measure of the household’s economic status.

Results’ section:

7) Generally, a final model should only include statistically significant variables, however this seems not to be true for what you reported as final model 4 – please clarify in the methods’ section how you arrived at the final models.

Discussion section:

8) Please revise the discussion section, and:

a) Include an introductory as the first paragraph for summarizing the study aim and all key findings.

b) Don’t repeat the methods and results’ sections and focus on interpreting the key findings as well as comparing them with the existing literature.

Reviewer #2: Please go through the manuscript to clean all grammatical errors, in addition to the comments attached. The manuscript in the current format has several grammatical errors that weaken in it in addition to the points raised in the attached document.

Reviewer #3: 1 Decision

After performing a careful read I recommend this paper for “acceptance” with some revisions and feel that the content is suitable for readers of the esteemed journal, PLOS Global Public Health conditional on the following comments/corrections being satisfied. The paper still has scope for improvement. Some paragraphs need to be clarified to improve readability. Please see below:

2 Major Comments

1. The paper starts with a long introduction. Please divide it into four/five paragraphs. Also, can the abstract be shortened. Being a little more specific and succinct could save a lot of real estate and shorten the paper, making it a better read.

2. The authors should clarify what exactly is their methodological contribution. It seems they are just using existing methodologies.

3. Ghana now has 16 regions in Ghana. I think considering the current restructuring will make the paper more appealing to the readers.

4. The use of citations in the paper is repetitive, it is generally considered good practice to be more specific in citations. For instance, under independent variables, you do need 10 references. Consider doing the same for the rest of the paper.

5. On Page 10 out of 36, consider citing as least this paper at least, Liu, Q., & Pierce, D. A. (1994). A note on Gauss—Hermite quadrature. Biometrika, 81(3), 624-629 and mostly explain why you decided to use this approach in your paper.

6. In Table 1, can you show both the numbers and percentages under health and non-healthy facility. It will be interesting to see those numbers.

7. I think it is better to have the strengths, limitations, and conclusion be under discussion, and provide some recommendations. Again, summarize in less and concise paragraphs.

3 Minor Comments

1. Page 3 out of 36, space before citing references.

2. First define SDG.

3. Page 4 out of 36, space before citing references and do same throughout the paper.

4. Check the citation under “ethics approval and consent to participate.”

Suggested References

Liu, Q., & Pierce, D. A. (1994). A note on Gauss—Hermite quadrature. Biometrika, 81(3), 624-629.

6. PLOS authors have the option to publish the peer review history of their article (what does this mean?). If published, this will include your full peer review and any attached files.

**Do you want your identity to be public for this peer review?** For information about this choice, including consent withdrawal, please see our Privacy Policy.

Reviewer #1: No

Reviewer #2: No

Reviewer #3: No

---

## [Editor Report · Decision Letter 1]

20 Jun 2023

PGPH-D-22-01485R1

A multilevel analysis of the predictors of health facility delivery in Ghana: evidence from the 2014 Demographic and Health Survey

Dear Dr. Aheto,

Thank you for submitting your manuscript to PLOS Global Public Health. After careful consideration, we feel that it has merit but does not fully meet PLOS Global Public Health’s publication criteria as it currently stands. Therefore, we invite you to submit a revised version of the manuscript that addresses the points raised during the review process.

We look forward to receiving your revised manuscript.

Kind regards,

Bethany Hedt-Gauthier, PhD

Academic Editor

Journal Requirements:

Additional Editor Comments (if provided):

Thank you for your resubmission; I have read your responses closely and reread the manuscript. Given the delays in reviews from the first round, I want to make sure this is as tight as possible before sending to reviewers again. As such, please see my comments below. I kindly ask that: 1) you reply to these comments on their own; 2) that you resend your response to reviewers but make any updates to your responses to reflect changes based on these comments as well. (The reviewers will review your responses in the next phase.)

1) I noted your replies about the datedness of the data and the reasoning for this. Please update in the manuscript that a) this is the most recent full DHS (presuming the 2017 DHS does not have sufficient sample size?); b) in your limitations, discuss why you think the results will or will not hold (or what results will and will not hold) the test of time? Will there be changes in the distributions in Table 1? Will the effect sizes of the independent variables be maintained? Is the influence of household or community variables still the same?

2) I am not entirely convinced by your response about the use of the terms "geospatial" (in the paper or in your response). Specifically, what you are doing is using the coordinates of the respondents to identify the region to then calculate proportions with facility delivery. I would say this is GIS to calculate regional proportions and not "geospatial mapping" - from the reviewer, and I agree, you are making this seem more detailed than it really is. It also isn't clear that this adds anything to the paper. You don't put these proportions into the model, right? Could you have calculated this on a more granular level and included? Personally, if you aren't doing more with this data, then I suggest you drop this section. If you do keep, you need to make it more clear what this adds to the paper, be more specific on the techniques, and talk about how obfuscated coordinates where considered in these estimates.

3) Your discussion is very long, and in its length hides the key points. You are arguing the novelty of the paper, as compared to the many other papers looking at predictors of facility delivery in Ghana, is the multilevel modeling. This is where your discussion emphasis should be, rather than over discussion/interpretation of the specific individual parameters. Can you give a larger percent representation (discuss more, and discuss the other sections less) on: a) What can we say about HH and community level factors given the results of the 4 models? (and for this point, in the conclusion "intervening on HH and community factors" needs more specifics that are introduced earlier - what interventions? can we say that now or is more work needed?) and b) How does what we say about individual factors change when we account for these nested structure of the data? (I would argue from your results, not much. But this is totally lost in the hyper-interpretation of these effect sizes)

4) As you are making these changes, make these minor edits as well:

- Please only use one number after the decimal (both to be consistent and to not overstate your precision). For example, on page 12, 2.99%->3.0%, but note this is in text and tables.

- I still find Table 1 difficult to draw conclusions. Can you change the column percents to be relative within columns (i.e. sum to 100% within each column).

- In conclusion, change cover to coverage.
---

## [Decision Letter · Decision Letter 2]

26 Oct 2023

PGPH-D-22-01485R2

A multilevel analysis of the predictors of health facility delivery in Ghana: evidence from the 2014 Demographic and Health Survey

Dear Dr. Aheto,

Thank you for submitting your manuscript to PLOS Global Public Health. After careful consideration, we feel that it has merit but does not fully meet PLOS Global Public Health’s publication criteria as it currently stands. Therefore, we invite you to submit a revised version of the manuscript that addresses the points raised during the review process.

We look forward to receiving your revised manuscript.

Kind regards,

Bethany Hedt-Gauthier, PhD

Academic Editor

Journal Requirements:

1. Please provide additional details regarding participant consent. In the ethics statement, please ensure that you have specified what type you obtained (for instance, written or verbal, and if verbal, how it was documented and witnessed). If your study included minors, state whether you obtained consent from parents or guardians.

Additional Editor Comments (if provided):

Overall, I and the reviewer found that you were responsive to the initial comments. We have some additional comments, mostly minor.

More substantive edits:

- Please walk back the statement: "... it guarantees a reduction in complications during and immediately after delivery, a safe birth, and maximises the survival chances..." This is an overstatement of what we know and what we can conclude about giving birth in facilities.

- Remove all of the mapping references, including the map itself. It really doesn't add, and in my opinion detracts from the paper.

- Your last sentence of Independent variables: "... but these were removed during model selection..."? Say more, why? What was the process? Your first table you show everything is significant, is this because you removed the insignificant factors? This needs to be clarified, and most likely moved to the analysis section.

- Results - you keep jumping tenses. This should probably be in past tense. More than anything, be consistent. (It is throughout the results, but for example, "We then extended model 1..."

- Doesn't Model 4 have household and community factors. Is it not possible to calculate the household-level VPC and MOR? (see table). Very possible this cannot be done, and if so, then be sure to add a footnote.

- One comment you did not address here - in the limitations, you say this is the most recent data. Do you think these results will hold the test of time? Still relevant now?

- One comment you did not address here - in the conclusion, you say "We therefore propose a household..." which households/communities? ANd what interventions? (I don't think you can say what interventions based on what is here, but maybe you can... if not, then you need to say this has to be studied.

Minor edits:

- In the introduction, change to STD to STG (STD target 3.7)

- Remove the commas in this sentence to read: "Though maternal and child healthcare and delivery services are free, ...."

- Add a comma to "These include maternal education (categorized as no education, primary....)

- Edit to ... nested within household which are nested...

-

Reviewers' comments:

Reviewer's Responses to Questions

**Comments to the Author**

1. If the authors have adequately addressed your comments raised in a previous round of review and you feel that this manuscript is now acceptable for publication, you may indicate that here to bypass the “Comments to the Author” section, enter your conflict of interest statement in the “Confidential to Editor” section, and submit your "Accept" recommendation.

Reviewer #3: All comments have been addressed

2. Does this manuscript meet PLOS Global Public Health’s publication criteria? Is the manuscript technically sound, and do the data support the conclusions? The manuscript must describe methodologically and ethically rigorous research with conclusions that are appropriately drawn based on the data presented.

Reviewer #3: Yes

3. Has the statistical analysis been performed appropriately and rigorously?

Reviewer #3: Yes

4. Have the authors made all data underlying the findings in their manuscript fully available (please refer to the Data Availability Statement at the start of the manuscript PDF file)?

Reviewer #3: Yes

5. Is the manuscript presented in an intelligible fashion and written in standard English?

Reviewer #3: Yes

6. Review Comments to the Author

Reviewer #3: 1) How did you select the independent variables? Need some clarifications.

2) How was missing data handled?

3) Space before citing references, see Page 4 of 31 among others.

4) For consistency it will be best to represent all confidence intervals as (0.2-0.5) or (0.2, 0.5). Just stick to one of these. Also, space out number (percentage) in Table 1.

7. PLOS authors have the option to publish the peer review history of their article (what does this mean?). If published, this will include your full peer review and any attached files.

**Do you want your identity to be public for this peer review?** For information about this choice, including consent withdrawal, please see our Privacy Policy.

Reviewer #3: No

---

## [Decision Letter · Decision Letter 3]

26 Feb 2024

A multilevel analysis of the predictors of health facility delivery in Ghana: evidence from the 2014 Demographic and Health Survey

PGPH-D-22-01485R3

Dear Dr. Aheto,

We are pleased to inform you that your manuscript 'A multilevel analysis of the predictors of health facility delivery in Ghana: evidence from the 2014 Demographic and Health Survey' has been provisionally accepted for publication in PLOS Global Public Health.

Best regards,

Julia Robinson

Executive Editor

Reviewer Comments (if any, and for reference):

Reviewer's Responses to Questions

**Comments to the Author**

1. If the authors have adequately addressed your comments raised in a previous round of review and you feel that this manuscript is now acceptable for publication, you may indicate that here to bypass the “Comments to the Author” section, enter your conflict of interest statement in the “Confidential to Editor” section, and submit your "Accept" recommendation.

Reviewer #3: All comments have been addressed

2. Does this manuscript meet PLOS Global Public Health’s publication criteria? Is the manuscript technically sound, and do the data support the conclusions? The manuscript must describe methodologically and ethically rigorous research with conclusions that are appropriately drawn based on the data presented.

Reviewer #3: Yes

3. Has the statistical analysis been performed appropriately and rigorously?

Reviewer #3: Yes

4. Have the authors made all data underlying the findings in their manuscript fully available (please refer to the Data Availability Statement at the start of the manuscript PDF file)?

Reviewer #3: Yes

5. Is the manuscript presented in an intelligible fashion and written in standard English?

Reviewer #3: Yes

6. Review Comments to the Author

Reviewer #3: (No Response)

7. PLOS authors have the option to publish the peer review history of their article (what does this mean?). If published, this will include your full peer review and any attached files.

**Do you want your identity to be public for this peer review?** For information about this choice, including consent withdrawal, please see our Privacy Policy.

Reviewer #3: No
